# A Neoteric Feature Extraction Technique to Predict the Survival of Gastric Cancer Patients

**DOI:** 10.3390/diagnostics14090954

**Published:** 2024-05-01

**Authors:** Warid Islam, Neman Abdoli, Tasfiq E. Alam, Meredith Jones, Bornface M. Mutembei, Feng Yan, Qinggong Tang

**Affiliations:** 1School of Electrical and Computer Engineering, University of Oklahoma, Norman, OK 73019, USA; warid9666@ou.edu (W.I.); neman.abdoli@ou.edu (N.A.); 2School of Industrial and Systems Engineering, University of Oklahoma, Norman, OK 73019, USA; tasfiq@ou.edu; 3Stephenson School of Biomedical Engineering, University of Oklahoma, Norman, OK 73019, USA; meredith.jones@ou.edu (M.J.); bmutembei@ou.edu (B.M.M.); feng.yan@ou.edu (F.Y.)

**Keywords:** gastric cancer, feature extraction, random forest (RF) classifier, radiomics features, support vector machine (SVM), K-Nearest Neighbors (KNN), Naïve Bayes (NB), area under ROC curve (AUC)

## Abstract

Background: At the time of cancer diagnosis, it is crucial to accurately classify malignant gastric tumors and the possibility that patients will survive. Objective: This study aims to investigate the feasibility of identifying and applying a new feature extraction technique to predict the survival of gastric cancer patients. Methods: A retrospective dataset including the computed tomography (CT) images of 135 patients was assembled. Among them, 68 patients survived longer than three years. Several sets of radiomics features were extracted and were incorporated into a machine learning model, and their classification performance was characterized. To improve the classification performance, we further extracted another 27 texture and roughness parameters with 2484 superficial and spatial features to propose a new feature pool. This new feature set was added into the machine learning model and its performance was analyzed. To determine the best model for our experiment, Random Forest (RF) classifier, Support Vector Machine (SVM), K-Nearest Neighbors (KNN), and Naïve Bayes (NB) (four of the most popular machine learning models) were utilized. The models were trained and tested using the five-fold cross-validation method. Results: Using the area under ROC curve (AUC) as an evaluation index, the model that was generated using the new feature pool yields AUC = 0.98 ± 0.01, which was significantly higher than the models created using the traditional radiomics feature set (*p* < 0.04). RF classifier performed better than the other machine learning models. Conclusions: This study demonstrated that although radiomics features produced good classification performance, creating new feature sets significantly improved the model performance.

## 1. Introduction

Gastric cancer (GC), one of the most prevalent malignant tumors, ranks as the second most common cause of cancer-related deaths worldwide [1]. Surgical procedures and subsequent chemotherapy are common therapeutic options. At the earliest possible stage of cancer diagnosis and prior to surgery, it is critical to accurately forecast the aggressiveness of malignant gastric tumors and the likelihood that patients will survive. Computed tomography (CT) is currently employed in clinical practice as the most common imaging modality to identify and diagnose gastric cancer [2]. We believe that CT scans can provide useful tumor phenotypic data related to tumor aggressiveness, which can be recognized and retrieved to assist in determining the chances of patients’ overall survival (OS).

The American Joint Committee on Cancer (AJCC) staging system is a clinically used guideline for treatment allocation and prognostication on gastric cancer patients [3,4,5]. However, there may still be significant differences in the actual survival results among patients with the same stage, suggesting that the current staging system is inadequate for individualized survival prediction [6]. Accurate survival prediction is difficult due to the high degree of heterogeneity in the pathophysiology of gastric cancer and the complicated etiologic and postoperative variables, but it is essential to improve patient treatment. Artificial intelligence (AI) has thus been extensively utilized in the healthcare sector recently to address this issue. AI refers to a machine’s ability to replicate intelligent human behavior [7], and it is clear that a digital metamorphosis is taking place in healthcare as the use of AI grows [8,9]. AI has proven its potential of using large datasets to perform specific tasks such as image recognition. Machine learning is an application of AI which refers to an analytical technique in which models are trained on databases to generate predictions on new, unforeseen data [10]. By examining medical images, the field of radiomics, a distinct branch of AI, extracts quantitative features using data-characterization algorithms [11]. The well-established radiomics concept suggests that some radiomics features computed from radiographic images such as CT images strongly correlate with cancer prognosis, and that using computer-aided detection (CAD) schemes can more effectively extract clinically relevant radiomics features to create machine-learning models to predict cancer prognoses [12]. For instance, in our previous studies, we have developed CAD-supported machine learning models to predict the classification of breast cancer patients [13] and the risk of cancer recurrence in lung cancer patients after surgery [14].

Advanced proximal gastric cancer patients can be predicted to develop hilar lymph node metastases using radiomic characteristics in conjunction with other clinical indicators [15]. Shin et al. achieved an AUC of 0.714 by evaluating the performance of a radiomic signature-based model for predicting recurrence free survival (RFS) of locally advanced gastric cancer using preoperative contrast-enhanced CT [16]. Pham et al. uses radiomics features to predict the survival rate of gastric cancer patients and achieved an AUC of 0.72 by assuming that no additional clinical information, such as, age, gender, surgery history, and location of residence, is available, thus solely evaluating radiomic characteristics as a tool for predicting survival rate [17]. However, none of the earlier investigations compared the impact of various additional features with the radiomics features on the final classification performance. As a result, developing a new feature set may be useful for enhancing the performance of the machine learning model.

Medical images consist of many input features. It is pertinent to reduce the high dimensionality of the features such that only a small number of informative features are selected for classification [18], thereby improving the performance and reducing the complexity of the model.

The reduced feature sets can be fed into the machine learning model to predict the probability of gastric cancer patients’ survival. Density-based methods are not effective for high dimensional data and usually require many training samples for a decent performance. This can be avoided by using kernels, as in Support Vector Machine (SVM)-based methods, or by artificially generated outliers [19]. One of the drawbacks of using this method is that sometimes excessive amounts of outliers are generated, which could lead to the inaccurate performance of the model. Randomized methods such as the RF classifier may be used to address this issue because they can both lower the quantity of outliers to generate and the size of the feature space in which the outliers are generated [20]. Aydadenta et al. applied machine learning techniques such as SVM, Artificial Neural Network (ANN), Naïve Bayes, k-Nearest Neighbor (kNN), and RF algorithm to image data such as colon, ovarian, central nervous, and lung to find out which algorithm works better [21]. It was found that the RF classifier outperforms SVM.

In this study, we extracted 27 texture and roughness parameters with 2484 superficial and spatial features to create a new feature pool. The unpaired t-student tests were used among the selected features and the features which are significantly different (*p* < 0.05) were selected. The selected features would then be fed in the RF classifier and its performance was further analyzed and compared with traditional radiomics features.

## 2. Materials and Dataset

### 2.1. Image Dataset

In this study, we assembled a retrospective image dataset from an existing database previously collected in our medical imaging laboratory, which has been used in our previous work [16]. A prior study provided the specifics of this image dataset, including patient demographics and clinical data as well as CT image scanning techniques and attributes [15]. In brief, this image dataset includes abdominal CT perfusion images acquired from 135 patients with pathologically confirmed gastric cancer. The patients were selected from the Fujian Medical University Union Hospital (Fuzhou, China) from January 2013 to December 2015 and January 2016 to December 2017, respectively. For each of these patients, three clinicopathological findings or markers obtained from the tumor resection specimen following surgery were also retrospectively recorded. The radiologists have determined and drawn a single region of interest (ROI) on the transverse imaging section of a tumor for each patient based on the response evaluation criteria in solid tumors (RECIST) standards currently utilized in clinical practice. This ROI shows the largest tumor size (or diameter). To pinpoint the locations of the gastric tumors, we used the radiologists’ labeled CT image slices as ground truth. Other clinical diagnostic results and treatment outcomes are also recorded and collected in this dataset. We categorize these 135 patients into two groups based on the three-year survival statistics: Survival, and Non-Survival. According to patient follow-up information, 68 of these patients lived, while the other 67 did not. The dimensions of the images were 512 × 512. The following steps are applied to develop and test the new CAD schemes based on a novel feature extraction technique to predict likelihood of patients’ survival in this study.

### 2.2. Image Preprocessing

The input image was filtered with a Gaussian filter to remove the unwanted noise, and histogram equalization was performed to improve the contrast of the image [8]. In addition, the tumor was segmented based on the annotations from radiologists. We placed a segmentation seed based on the center of the tumor in one CT slice. On this CT slice, the tumor was segmented throughout the center region of the tumor from the seed location using an adaptive multi-layer region growth method [22]. A threshold value for the first layer was determined using the digital value of the local minimum point found in the region growing algorithm’s seed. This allowed for a partial correction of the Hurter Driffield (H-D) curve effect in the source films. Two additional threshold values were calculated for the second and third layers from the old threshold value and the contrast value. The detail of this process is described in [22]. We used MATLAB (Version: R2021a, Natick, MA, USA) to segment the tumor regions. Similarly, this algorithm has also been used for segmentation purposes in our previous studies [23,24]. The segmented tumors were visually reviewed and manually corrected in cases where the automatic segmented didn’t produce desirable results. Figure 1 shows the illustration of the segmentation process. Some more illustrations of the original images and the segmentation process are mentioned in Appendix A.

### 2.3. Feature Extraction

The pyradiomics platform is an open-source package developed to extract radiomics features from medical images, and the features extracted by this platform are in accordance with the feature definitions as described by the Imaging Biomarker Standardization Initiative (IBSI) [25]. In this research, we used pyradiomics platform [26] to extract radiomics features and calculate 73 features. The features are described below:

#### 2.3.1. Gray-Level Co-Occurrence Matrix (GLCM)

GLCM is a well-known statistical technique for extracting texture information from gray level images [27]. It depicts how the different shades of gray are distributed spatially and how they interact in each area. The GLCM approach can be used to locate an extremely gray-level pixel [28]. Rashed et al. illustrates the GLCM features with descriptions and equations.

#### 2.3.2. Gray-Level Run-Length Matrix (GLRLM)

GLRLM is a type of 2D histogram-like matrix that logs the occurrence of every possible combination of gray-level values and runs in a ROI for a specific direction. In the matrix, row and column keys are typically used to represent gray-level values and runs. The (i,j)-th item in the matrix indicates the number of pairings whose gray-level value is *i* and whose run length is *j*. Gray-level values and runs are often represented as row and column keys in the matrix [29,30]. Rashed et al. provide descriptions and equations for the GLRLM features [31].

#### 2.3.3. Shape Features

Connected sections in binary images that are invariant to translation, rotation, and scaling are a hallmark of MI. The shapes of objects can be demonstrated using the moments. The moment-invariant features space has multiple dimensions, which invariance acknowledges. The central moments, which determine the seven shape qualities, are independent of the object’s scale, orientation, or translation. Using the center of gravity, invariant moments at the origins of the central moments are translated [32]. Bharti et al. show the calculation of the moment invariance of order (p,q) [33]. In case of central invariance, the central moments of order (p,q) is calculated as shown in [34].

#### 2.3.4. First Order Texture Features

Several first order statistical features were computed including Energy, Entropy, Interquartile Range, Kurtosis, Maximum, Mean Absolute Deviation, Mean, Minimum, Range, Robust Mean Absolute Deviation, Root Mean Square, Skewness, Total Energy, Uniformity, and Variance.

#### 2.3.5. Gray Level Dependence Matrix (GLDM)

A Gray Level Dependence Matrix (GLDM) measures the dependencies between gray levels in an image. The detailed equation is described in this study [35].

During the feature extraction stage, we extracted a set of features from images that indicate significant information necessary for classification and diagnosis. The feature sets included texture-based features, shape-based features, and features calculated from the GLC) in four directions (0, 45, 90, and 135), the GLRLM in four directions (0, 45, 90, and 135), and the GLDM, respectively. Thus, we obtained 16 features from the GLCM method, 16 features from the GLRLM method, 14 features from the GLDM method, 12 features from the shape-based method, and 15 features from the first order texture-based features. A radiomics feature set was created using the features above.

Several other techniques like roughness [36], morphology [37], and multiscale [38] texture studies, can be utilized to extract surface and spatial properties from intensity images for pattern recognition and image classification, increasing the efficacy of structural image analysis. Characteristic distributions and patterns that are not immediately visible in the original intensity structures can be revealed using this spatial and superficial information from photos. Applications for spatial and surface feature analysis are numerous in satellite and geological image analysis, and are useful for deriving data on the distribution of regions, item identification, geological structures, and tectonics [39,40,41]. CT images combined with multiple texture analyses have been shown in some studies to be useful for tissue differentiation in diagnostics [42], retinal thickness classification [43], urinary bladder cancer recognition [44], ovarian tissue characterization [45], gastrointestinal tissues classification [46], oral carcinoma identification [47], and skin layer segmentation and surface quantification [48]. These benefits stem from the ability to expand image features through superficial and spatial parameters.

These additional features were extracted to find an effective feature pool. They are as follows:Statistical texture parameters: Histogram, Local Binary Descriptors (LBP), Histogram of Oriented Gradients (HOG), Gray Level Difference Statistics (GLDS), First-Order Statistics (FOS), Correlogram, Statistical Feature Matrices (SFM), and Gray Level Size Zone Matrix (GLSZM)One structural texture parameter: Shape-parameterOne model-based texture parameter: Fractal Dimension Texture Analysis (FDTA)Six transform-based texture parameters: Gabor Pattern (GP), Wavelet Packets (WP), Discrete Wavelet Transform (DWT), Stroke Width Transform (SWT), Higher Order Spectra (HOS), and Laws Texture Energy (LTE)One model-based texture parameter: Amplitude Modulation-Frequency Modulation (AMFM)Two image moments texture parameters: Zero moments and Hu momentsTwo pattern spectrum and shape size texture parameters: Multilevel Binary Morphological Analysis (MultiBNA) and Gray Scale Morphological Analysis (GSMA)One thresholding adjacency statistics (TAS) texture parameterOne multi-regional histogram (multiregional)One surface roughness parameter

The details of these parameters included: (1) Histogram (32 features), (2) Gray-Level Co-occurrence Matrix (GLCM, 28 features), (3) Local Binary Descriptors (LBP, 6 features), (4) Gray-Level Run Length Matrix (GLRLM, 10 features), (5) Histogram of Oriented Gradients (HOG, 4 features), (6) Gray Level Difference Statistics (GLDS, 5 features), (7) First-Order Statistics (FOS, 16 features), (8) Correlogram_ht (1024 features), (9) Correlogram_hd (1024 features), (10) Statistical Feature Matrices (SFM, 4 features), (11) Gray Level Size Zone Matrix (GLSZM, 14 features), (12) Shape-parameter (5 features), (13) Fractal Dimension Texture Analysis (FDTA, 6 features), (14) Gabor Pattern (GP, 4 features), (15) Wavelet Packets (WP, 126 features), (16) Discrete Wavelet Transform (DWT, 18 features), (17) Stroke Width Transform (SWT, 18 features), (18) Higher Order Spectra (HOS, 5 features), (19) Laws Texture Energy (LTE, 6 features), (20) Amplitude Modulation- Frequency Modulation (AMFM, 4 features), (21) Zero moments (25 features), (22) Hu moments (7 features), (23) Multilevel Binary Morphological Analysis (MultiBNA, 6 features), (24) Gray Scale Morphological Analysis (GSMA, 2 features), (25) thresholding adjacency statistics (TAS, 54 features), (26) multiregional (4 features), and (27) Roughness (26 features) parameters.

These additional features were used to create a new feature set. We used a total of 27 texture and roughness parameters to extract 2484 superficial and spatial features from each image. Figure 2 shows the schematic of our novel feature extraction method.

### 2.4. Random Forest Classifier

The RF algorithm is an extension of the bagging method which uses feature randomness in addition to bagging to produce an uncorrelated forest of decision trees [49]. The three primary hyperparameters of random forest algorithms must be set prior to training. The number of trees, node size, and sampled feature counts are a few of them. Following that, regression or classification issues can be resolved using the random forest classifier. Each decision tree in the ensemble that makes up the random forest method is built of a data sample taken from a training set with replacement known as the bootstrap sample. One-third of the training sample, or the out-of-bag (OOB) sample, is set aside as test data. The next step is feature bagging, which introduces yet another randomization event while increasing dataset variety and decreasing decision tree correlation. The method for making a prediction will differ depending on the type of problem. The individual decision trees will be averaged for the regression job, whereas for the classification task, the predicted class will be determined by the majority vote, or the most common categorical variable. The prediction is then finalized by cross-validation using the OOB sample. Figure 3 shows the schematic of the RF algorithm process. Some of the benefits of random forest classifier include:Reduced risk of overfitting, as the presence of robust numbers of decision trees reduces the overall variance and prediction error.Provides flexibility, as it can perform both regression and classification tasks with a high level of accuracy. The model is useful for estimating missing values due to the inclusion of feature bagging because it retains accuracy even when some of the data is missing.Easy to determine feature importance by evaluating the importance of a variable’s contribution to the model is simple with RF.

### 2.5. Model Preparation

After processing images and computing features from all 135 images, we built machine learning models to predict the survival of the patients. RF classifier was used as the machine learning model. The parameters used in RF classifier are Number of Trees (n_estimators), Maximum Depth of Trees (max-depth), Minimum Samples Split (min_samples_split), Minimum Samples Leaf (min_samples_leaf), Maximum Features (max_features), Criterion (criterion), and Bootstrap Samples (bootstrap). We performed hyperparameter tuning separately when comparing different feature spaces. This technique involves grid search or randomized search to find the optimal combination of hyperparameters for each feature space. Second, we used the unpaired t-student tests to select the features which are significantly different (*p* < 0.05). Third, to increase the size and diversity of the training cases, as well as reduce potential bias in case partition and selection, we used the five-fold cross validation to train the classification model and evaluate its performance [50]. After training, for each testing image, the machine learning model will generate a classification score to divide all the testing cases into two classes of Survival and Non-Survival. The score ranges from 0 to 1. From the numbers of testing cases in two classes, we computed and created confusion matrix by applying an operation threshold on the RF generated classification scores (T = 0.5). From the confusion matrices, the classification accuracy, sensitivity, specificity, and F1 score were computed and compared. Figure 4 shows the schematic of our model.

### 2.6. Performance Evaluation

We applied the following two statistical data analysis methods to evaluate and compare the gastric tumor classification performance of machine learning models based on different feature extraction techniques. First, we used a receiver operating characteristic (ROC) type data analysis method and the area under the ROC curve (AUC) as an evaluation index. Second, based on model-generated probability scores between two output/classification nodes that predicted patient survival (Survival/Non-Survival), a testing image was assigned to one class of either Survival or Non-Survival, based on which probability score was greater between two output nodes of the model. Then, a confusion matrix was generated for each testing subset. Based on the confusion matrix, several additional evaluation indices were computed, which included the overall classification accuracy, sensitivity, specificity, and F1 score computed as follows to evaluate model performances used in a binary or practical decision environment:Accuracy=(TP+TN)/(TP+TN+FP+FN)
Sensitivity=TP/(TP+FN)
Specificity=TN/(TN+FP)
F1 Score=2TP/(2TP+FP+FN)
where TP and TN represent the number of correctly classified Survival and Non-Survival cases, while FP and FN represent misclassified Survival and Non-Survival cases.

## 3. Results

We employed a total of 27 texture and roughness parameters to extract 2484 superficial and spatial features from each image. Within this pool of features, we applied the unpaired t-student tests among all the features. Figure 5 displays a heatmap of *p*-values less than 0.05 across all the features. Only the features that are significantly different (*p* < 0.05) were selected in our final model.

We exhibited the difference distribution of all the selected features. We used volcano plots to assess the significance and magnitude of the differences (Figure 6) as they are particularly effective for visualization. They plot the log2fold change on the x-axis (which shows the magnitude of the change) against the -log10 of the *p*-value on the y-axis (which indicates the statistical significance of the change). This helps to illustrate both a large magnitude of change and high statistical change. These are usually the most important features.

The selected features were fed into the RF classifier and their classification performance was analyzed. Another machine learning model was built using the traditional radiomics feature set. The two models were analyzed by computing several evaluation metrices. The AUC value of the model that was built using the new feature set is 0.98 ± 0.01 which is significantly higher than the AUC value (0.72 ± 0.03) of the model built using the traditional radiomics feature set (*p* < 0.05). The results are illustrated in Figure 7.

Secondly, based on the above confusion matrices in Figure 8, the computed classification accuracy of the model built using the new feature set is 0.92, which is higher than the model built using the conventional radiomics feature set (0.64). Additionally, sensitivity, specificity, and the F-1 score of the model built using the new feature set was higher than the machine learning model built using the traditional radiomics feature set as shown in Table 1.

We added the new feature pool to the other machine learning models such as SVM, KNN, and Naïve Bayes, and discovered that the RF model performs better than the other models. Table 2 displays the results we obtained.

## 4. Discussion

Different types of machine learning technologies have been extensively used to aid in the diagnosis of gastric cancer and prediction of patients’ prognosis or survival [51,52,53,54,55,56,57]. Feature extraction is an important part of the classification process as it enables the model to classify the images based on their distinct features. In this study, we investigated and evaluated a novel method to develop and optimize CAD schemes in the diagnosis of gastric cancer patients. The study has several unique characteristics and new observations. First, conventional CAD schemes train machine learning classifiers using radiomics feature sets. Wang et al. used similar images from 515 patients to develop a CT-based radiomics nomogram for identification of No.10 LNs status in advanced proximal gastric cancer (APGC) patients. They computed 352 image features from each segmented image. However, the performance of these models is still not impressive. In this study, we computed several other features. We tested a novel subset of features that yielded much higher classification performance in comparison to the traditional radiomics features as shown in the studies by Shin et al. and Pham et al. Texture and roughness feature extractions are widely used in remote sensing and materials to describe the intrinsic homogeneity/heterogeneity, scales, regularity, directionality, and isotropy/anisotropy. These features can extract specific structure information for image segmentation, classification, pattern recognition, and profile analysis. Texture and roughness features allow us to extract the intrinsic distribution of internal tissue structures. In the pursuit of extracting image features, various texture extraction methods have emerged, encompassing statistical, structural, model-based, and transform-based approaches within the machine vision field. These approaches focus on initial units, spatial layout, physical model designs, and transformation functions. Each method employs distinct mathematical models for extracting image features. Currently, there are 26 classic texture methods specifically developed for texture feature extraction [58]. Additionally, roughness, a parameter comprising 26 features, is employed to extract spatial information from the material surface [59]. These selected 27 texture and roughness parameters are broadly utilized in the analysis of remote sensing and material processing. Therefore, we applied these 27 parameters that can extract 2484 surficial and spatial features in total in our CT images. This indicated the importance of research effort to continuously search for novel and more effective model finetuning and optimization methods in the development of machine learning-based CAD schemes with respect to medical images. To the best of our knowledge, no similar CAD study has been reported in earlier literature.

Secondly, one of the popular machine learning models for classification is the Support Vector Machine (SVM). However, SVM, might not be efficient for all datasets. Our study showed that RF classifier performs better than SVM for our dataset. We also applied the K-Nearest Neighbors (KNN) and the Naïve Bayes model to our dataset and found out that the RF classifier outperformed all the other models.

Third, we did not have a large dataset for our experiment. If we used the traditional train-test split, we would face a smaller test set which could affect the result. To alleviate this problem, we used the five-fold cross validation technique so that we were able to make predictions on all our data. To assess the scientific rigor or robustness of our model, we repeated the experiment five times. The analysis results showed small standard deviations (i.e., <0.02), which demonstrated the robustness of the model.

Therefore, the study outcomes validate our hypothesis that using the new feature set yields better classification performance than the traditional radiomics features.

Despite the encouraging results or observations made in this study, we also recognize that the study has some limitations. We used a small dataset for our experiment (n = 135). To assess the experiment’s viability, a bigger dataset might be used. In addition, our study results also provide a new scientific rationale and background for us to develop deep learning model-based CAD schemes further optimally [36]. Instead of using handcrafted features, we could use deep learning to extract additional features. We could fuse the traditional handcrafted and the deep learning features and create a new feature pool. This new feature subset could be used to build a new machine learning model and its performance could be analyzed. Autoencoder neural network has been recently used extensively in medical image processing [60,61,62]. We plan to apply these networks to our dataset to see if we can come up with a better performance. The inference time on NVIDIA RTX 4090 GPUs was 0.19 s for this entire dataset. This timing could be reduced if multiple GPUs are available, and we plan to study them in our future work. Thus, the above limitations need to be further investigated in our future studies. However, we think that this study is reliable and that it successfully tests or validates the hypothesis that using the novel feature set extracted earlier to create a new feature pool would improve the performance of the machine learning model.

## 5. Conclusions

In this study, we proposed and investigated the feasibility of a new feature pool based on a new subset of features (27 texture and roughness parameters). The study’s findings showed that the new feature set outperformed the traditional radiomics feature set in terms of performance. The robustness of the method was also tested and demonstrated by repeating the experiment five times. This provides a solid platform to further perform this experiment on a larger dataset and combine it with deep learning to produce better results.

## Figures and Tables

**Figure 1 diagnostics-14-00954-f001:**
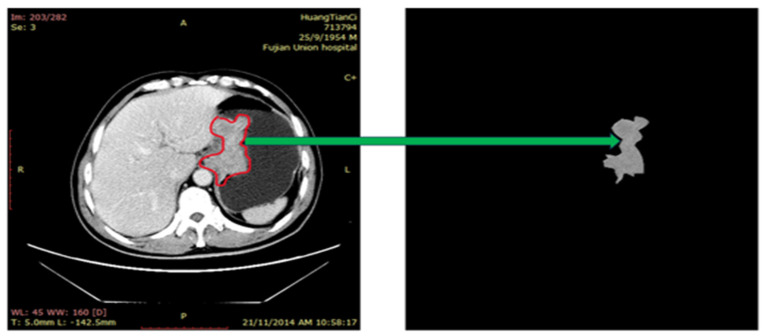
Example of illustrating segmented tumor regions on CT slice.

**Figure 2 diagnostics-14-00954-f002:**
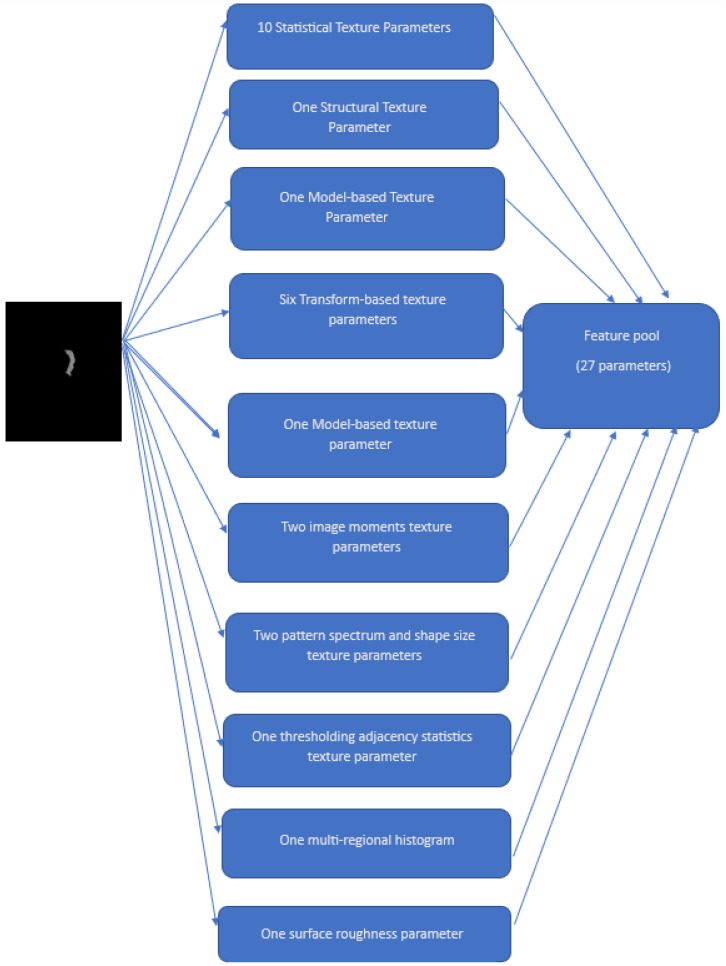
Flow diagram of feature extraction process.

**Figure 3 diagnostics-14-00954-f003:**
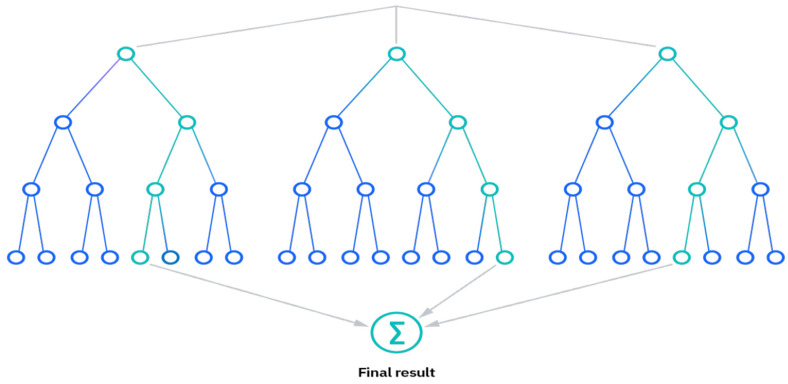
Illustration of RF Model.

**Figure 4 diagnostics-14-00954-f004:**
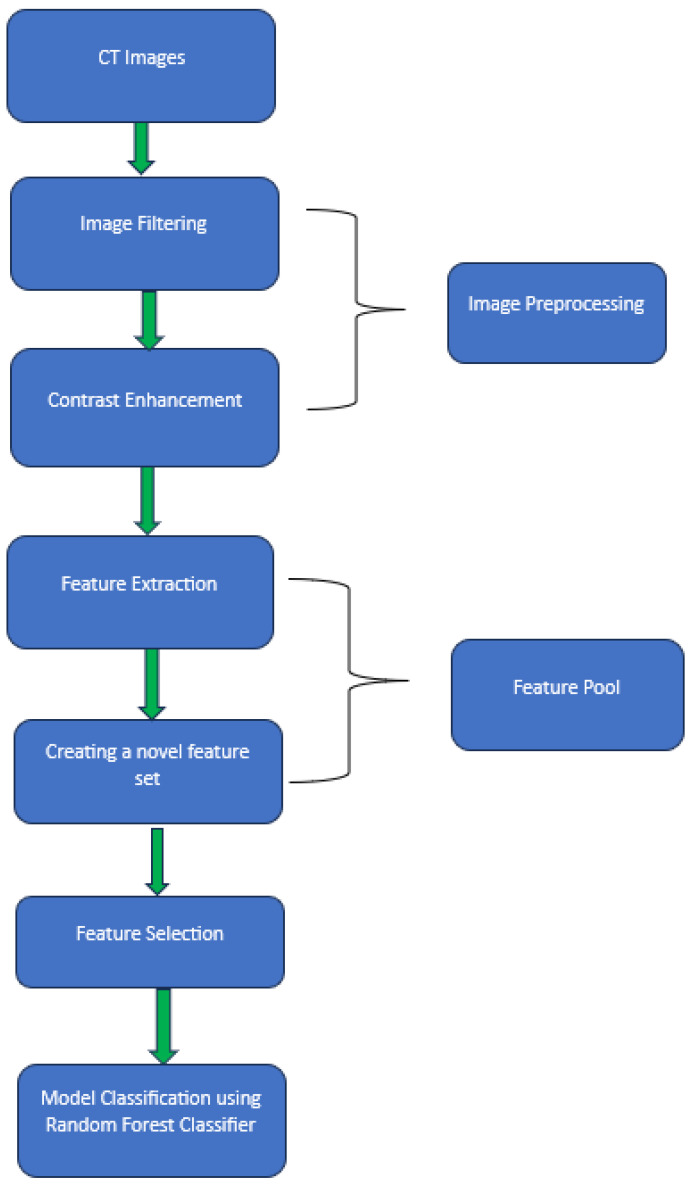
Schematic of machine learning model.

**Figure 5 diagnostics-14-00954-f005:**
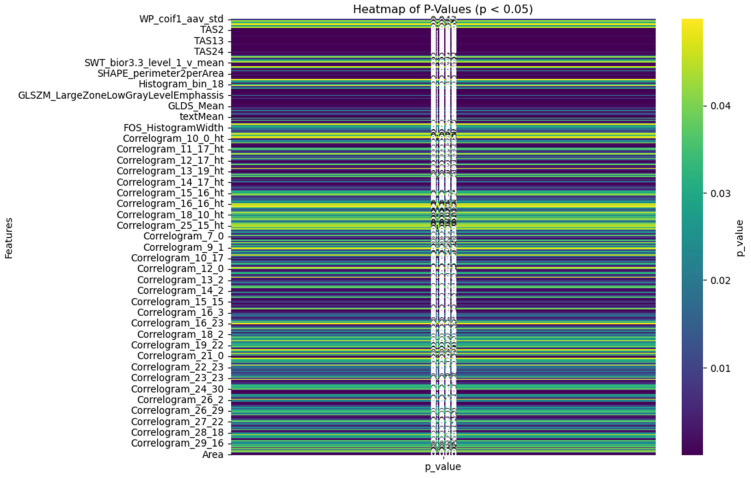
Heatmap of features with *p*-values less than 0.05.

**Figure 6 diagnostics-14-00954-f006:**
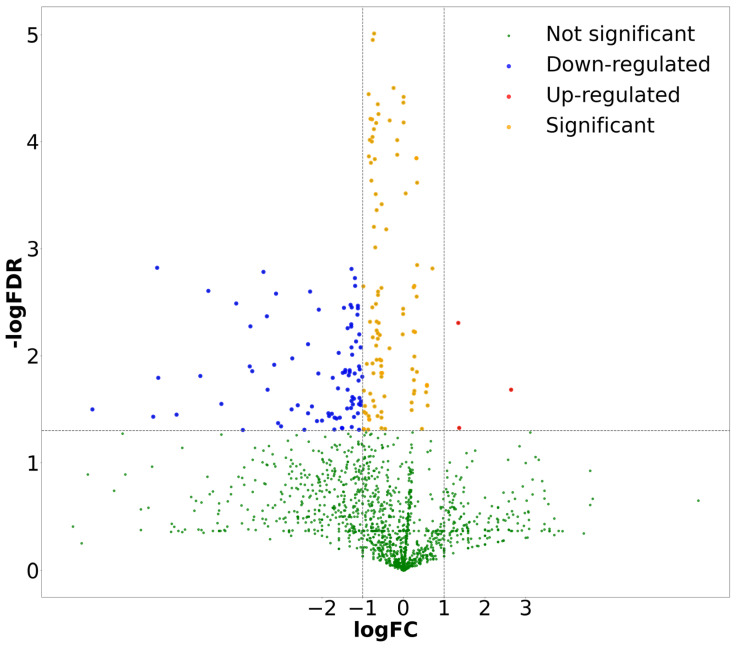
Volcano plot of difference among selected features.

**Figure 7 diagnostics-14-00954-f007:**
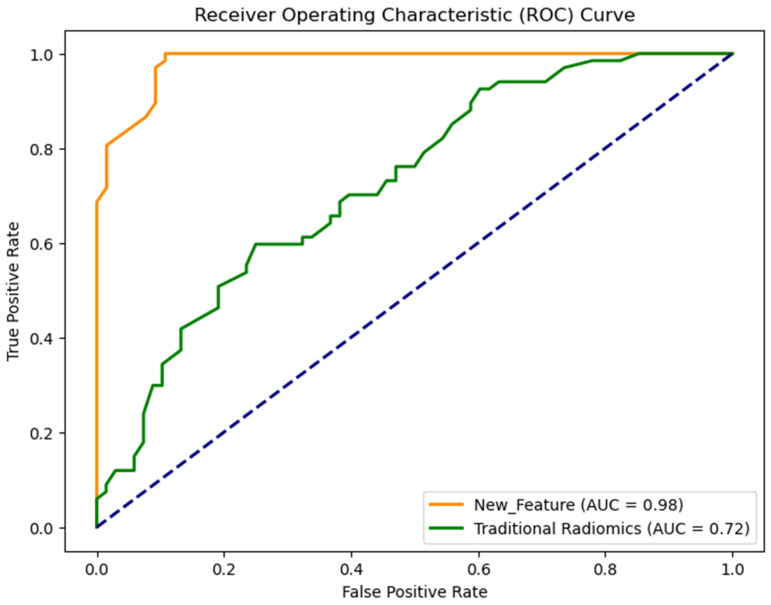
Receiver operating characteristic (ROC) curve of two models.

**Figure 8 diagnostics-14-00954-f008:**
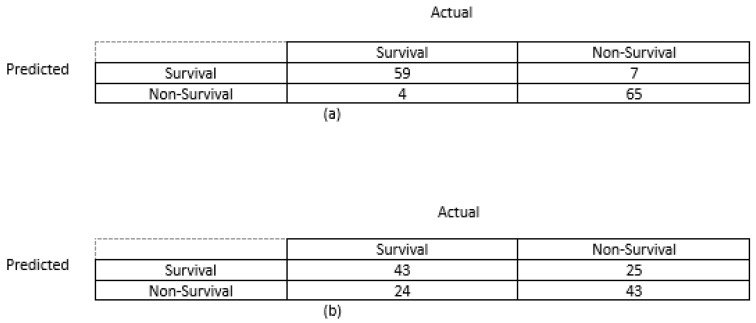
(**a**) Confusion matrix generated from model built using new feature set after five-fold cross-validation across entire dataset. (**b**) Confusion matrix generated from model built using traditional radiomics feature set after five-fold cross validation across entire dataset.

**Table 1 diagnostics-14-00954-t001:** Performance evaluation of the RF model using the two feature sets.

Feature Set	AUC	Accuracy	Sensitivity	Specificity	F-1 Score
Radiomics	0.72 ± 0.02	0.64 ± 0.04	0.64 ± 0.04	0.63 ± 0.02	0.64 ± 0.03
New	0.98 ± 0.01	0.92 ± 0.02	0.94 ± 0.03	0.90 ± 0.03	0.92 ± 0.04

**Table 2 diagnostics-14-00954-t002:** Performance comparison of two machine learning models using new feature set.

Machine Learning Model	Accuracy	AUC
SVM	0.90	0.93
Random Forest Classifier	0.92	0.98
KNN	0.91	0.93
Naïve Bayes	0.91	0.94

## Data Availability

The data presented in this study are available on request from the corresponding author (qtang@ou.edu).

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
