# Peer review of "A Neoteric Feature Extraction Technique to Predict the Survival of Gastric Cancer Patients"

_diagnostics, 2024, doi:10.3390/diagnostics14090954_

Round 1
Reviewer 1 Report
Comments and Suggestions for Authors
I would appreciate the authors for the prepared paper. Please consider the following comments.
1- The quality and size of Figure 6 is not good.
2- Provide more descriptions about the dataset. How did you get this dataset? Is it available in the internet? Depict at least 20 images of it. What is the class of each figure? How did assign the class of image (I mean the target)? Are there any other researches on this dataset? What is the dimension of figures? Provide a subsection about the dataset and describe all these points.
3- Describe clearly how did you segment your image (subsection 2.2)? Did you use a python ready package? Which software did you use? Depict the results at least for ten images in your dataset.
4- The literature review is very weak. Provide a more comprehensive literature review about Gastric Cancer prediction with machine learning using images. For example, cite https://www.sciencedirect.com/science/article/pii/S0933365722001762. Add a literature table for the papers.
5- Add some descriptions about using of auto encoders in health prediction. You can cite the following papers:
https://www.nature.com/articles/s41598-021-83184-2
https://www.sciencedirect.com/science/article/pii/S0933365723002038?via%3Dihub
https://www.ncbi.nlm.nih.gov/pmc/articles/PMC9148960/
Comments on the Quality of English Language
Good
Reviewer 2 Report
Comments and Suggestions for Authors
Dear authors, the presented manuscript is valuable for the scientific community in the area of medical imaging. Data processing and feature extraction have the same importance as machine learning algorithm optimization. Therefore, new approaches development is interesting and especially promising in case of small datasets such as in medical tasks. Here are my suggestions and comments:
1) I suggest providing more details about feature representation (converting 27 parameters to 2484 features).
2) When you compare different feature spaces, which parameters of the RF classifier do you use? Are they the same? It is assumed that the optimal number of trees, depth, and other parameters can vary depending on the feature set.
3) I suggest adding more explanations and conclusions from Figure 6.
4) It might be reasonable to mention in the caption for Figure 8 that the results are presented from 5-fold cross validation and cover the entire dataset.
5) I suggest adding more discussion about the exact new features' importance. Which one and why are the most significant? How do they help to increase the explainability of ML-based image classification?
Overall, I think your manuscript has great potential and these suggestions will help improve its clarity and impact.
Reviewer 3 Report
Comments and Suggestions for Authors
1. The novelty of the is poor. How can adding extra features improve the classification result?
2. Related works should be greatly improved in support of the work presented.
3. "However, none of the earlier investigations compared the impact of various additional features with the radiomics features on the final classification performance". Can the authors list out the radiomics features of previous works?
4. Whether the authors compared the complexity of the proposed model with previous works?
5. Why did the authors choose only SVM and random forest classifiers? Whether the proposed model is suitable for other classifiers like KNN, Naive Bayes? Table 2 can be improved with additional ML Classifiers.
6. The importance of the additional 27 features must be included. So that readers can understand why these additional features play a great boost in performance evaluation as listed in Table 1.
Overall, the authors a find gap in the analysis but failed to explain it clearly and meaningful pieces of information are missing.
